# Spin-liquid-like state in a spin-1/2 square-lattice antiferromagnet perovskite induced by $d^{10}$–$d^0$ cation mixing

O. Mustonen [1], S. Vasala [2], E. Sadrollahi[3], K.P. Schmidt[3], C. Baines[4], H.C. Walker [5], I. Terasaki [6], F.J. Litterst[2,3], E. Baggio-Saitovitch[2] & M. Karppinen [1]

A quantum spin liquid state has long been predicted to arise in spin-1/2 Heisenberg square-lattice antiferromagnets at the boundary region between Néel (nearest-neighbor interaction dominates) and columnar (next-nearest-neighbor interaction dominates) antiferromagnetic order. However, there are no known compounds in this region. Here we use $d^{10}$–$d^0$ cation mixing to tune the magnetic interactions on the square lattice while simultaneously introducing disorder. We find spin-liquid-like behavior in the double perovskite $Sr_2Cu(Te_{0.5}W_{0.5})O_6$, where the isostructural end phases $Sr_2CuTeO_6$ and $Sr_2CuWO_6$ are Néel and columnar type antiferromagnets, respectively. We show that magnetism in $Sr_2Cu(Te_{0.5}W_{0.5})O_6$ is entirely dynamic down to 19 mK. Additionally, we observe at low temperatures for $Sr_2Cu(Te_{0.5}W_{0.5})O_6$—similar to several spin liquid candidates—a plateau in muon spin relaxation rate and a strong $T$-linear dependence in specific heat. Our observations for $Sr_2Cu(Te_{0.5}W_{0.5})O_6$ highlight the role of disorder in addition to magnetic frustration in spin liquid physics.

[1] Department of Chemistry and Materials Science, Aalto University, FI-00076 Espoo, Finland. [2] Centro Brasileiro de Pesquisas Físicas (CBPF), Rua Dr Xavier Sigaud 150, Urca, Rio de Janeiro 22290-180, Brazil. [3] Institut für Physik der Kondensierten Materie, Technische Universität Braunschweig, 38110 Braunschweig, Germany. [4] Laboratory for Muon Spin Spectroscopy, Paul Scherrer Institut, 5232 Villigen PSI, Switzerland. [5] ISIS Neutron and Muon Source, Rutherford Appleton Laboratory, Chilton, Didcot OX11 0QX, UK. [6] Department of Physics, Nagoya University, Nagoya 464-8602, Japan. Correspondence and requests for materials should be addressed to M.K. (email: maarit.karppinen@aalto.fi)

Antiferromagnetic interactions on simple geometric lattices, such as triangular, square or tetrahedral, can give rise to magnetic frustration, because not all interactions between neighboring spins can be satisfied. These frustrated magnets have been widely studied in the search for exotic ground states such as quantum spin liquid (QSL) and quantum spin ice[1]. The square lattice has been of special interest due to its connection to high-temperature superconductivity[2]. Frustrated magnetism on a square lattice can be described using the spin-1/2 Heisenberg square-lattice model ($J_1$–$J_2$ model). This model has two interactions: nearest-neighbor interaction $J_1$ along the side of the square and next-nearest-neighbor interaction $J_2$ along the diagonal of the square (Fig. 1a). The $J_1$–$J_2$ model has three classical ground states: ferromagnetic (FM), Néel antiferromagnetic (NAF) and columnar antiferromagnetic (CAF) order. The Néel order occurs when the $J_1$ interaction is antiferromagnetic and dominates ($J_2/J_1 \ll 0.5$), while the columnar order requires a dominant antiferromagnetic $J_2$ interaction ($J_2/J_1 \gg 0.5$)[3].

The nature of the ground state in the highly frustrated region at the NAF–CAF boundary near $J_2/J_1 \approx 0.5$ is under debate. Anderson[4] famously proposed that a QSL state emerges when Néel order is frustrated by including an antiferromagnetic $J_2$ interaction. Quantum spin liquids are highly entangled states, in which spins remain dynamic even at absolute zero[1,5]. Experimental QSL candidates are known with several different structure types[6–11], typically Kagomé lattices, but a square-lattice QSL has not been realized. The other ground state suggested for the $J_2/J_1 \approx 0.5$ region is a valence bond solid[12–14], in which spins form dimer or plaquette singlets with a static pattern. Despite these theoretical predictions for the square-lattice antiferromagnets, no experimental evidence of a compound in the $J_2/J_1 \approx 0.5$ region exists.

Isostructural $A_2B'B''O_6$ double perovskite antiferromagnets $Sr_2CuTeO_6$ and $Sr_2CuWO_6$, where $A = Sr^{2+}$, $B' = Cu^{2+}$ and $B'' = Te^{6+}/W^{6+}$ (Fig. 1b), are well described by the $J_1$–$J_2$ model[15–20]. A Jahn–Teller distortion and an accompanying orbital ordering result in a square lattice of $Cu^{2+}$ ($S = 1/2$) ions with highly two-dimensional magnetic interactions[15,21]. The two $B''$ cations, $Te^{6+}$ and $W^{6+}$, have nearly the same size[22], and thus the bond distances and angles in $Sr_2CuTeO_6$ and $Sr_2CuWO_6$ are very similar[21]. Nevertheless, the diamagnetic $B''$ cation has a significant effect on the magnetic properties. Recent neutron scattering studies have revealed NAF ordering at $T_N = 29$ K with $J_1 = -7.18$ and $J_2 = -0.21$ meV ($J_2/J_1 = 0.03$) for $Sr_2CuTeO_6$, whereas $Sr_2CuWO_6$ has CAF ordering at $T_N = 24$ K with $J_1 = -1.2$ and $J_2 = -9.5$ meV ($J_2/J_1 = 7.92$)[18,19,23,24] (Fig. 1c). This dramatic change in exchange interactions is driven by differences in orbital hybridization. In $Sr_2CuWO_6$, the dominant 180° Cu–O–W–O–Cu $J_2$ exchange pathway is enabled by significant W $5d^0$–O $2p$ hybridization[19,25]. In contrast, the filled $4d^{10}$ states in $Sr_2CuTeO_6$ are core-like and do not hybridize[18,25], resulting in a weak $J_2$. The origin of the dominant 90° $J_1$ interaction is under debate: Babkevich et al.[18] proposed a Cu–O–O–Cu exchange pathway without a contribution from Te, whereas Xu et al.[25] proposed that some Te $5p$–O $2p$ hybridization does occur affecting the $J_1$ interaction. Since $Sr_2CuTeO_6$ has a dominant $J_1$ interaction and $Sr_2CuWO_6$ a dominant $J_2$, it is natural to ask whether the $J_2/J_1 \approx 0.5$ region could be reached by making a $Sr_2Cu(Te_{1-x}W_x)O_6$ solid solution.

Recently, Zhu et al.[26] showed that $Te^{6+}$–$W^{6+}$ ($d^{10}$–$d^0$) cation mixing can be used to tune the magnetic ground state of $Cr^{3+}$ ($S = 3/2$) inverse trirutiles $Cr_2TeO_6$ and $Cr_2WO_6$. Similar to $Sr_2CuWO_6$, W $5d^0$–O $2p$ hybridization in $Cr_2WO_6$ allows an exchange pathway not observed in $Cr_2TeO_6$, resulting in different magnetic structures for the two compounds. Magnetic interactions can be tuned by making a $Cr_2(Te_{1-x}W_x)O_6$ solid solution; a change in magnetic structure occurs at $x = 0.7$. Differences in the magnetic properties of isostructural $d^{10}$ and $d^0$ compounds have also been observed in perovskite-like $Ni^{2+}$ ($S = 1$)[27,28] and $Os^{6+}$ ($S = 1$)[29] compounds.

Here we show that the magnetic ground state of a spin-1/2 square-lattice antiferromagnet can be tuned by $d^{10}$–$d^0$ cation mixing. In the solid solution $Sr_2Cu(Te_{0.5}W_{0.5})O_6$ spins remain entirely dynamic down to 19 mK. This represents a suppression of $T_N$ by at least three orders of magnitude compared to the antiferromagnetic parent phases. Moreover, the magnetic specific heat shows $T$-linear behavior at low temperatures, despite the material itself being an insulator. These results indicate a spin-liquid-like ground state. A special property of $Sr_2Cu(Te_{0.5}W_{0.5})O_6$ is the high amount of quenched disorder in the magnetic interactions.

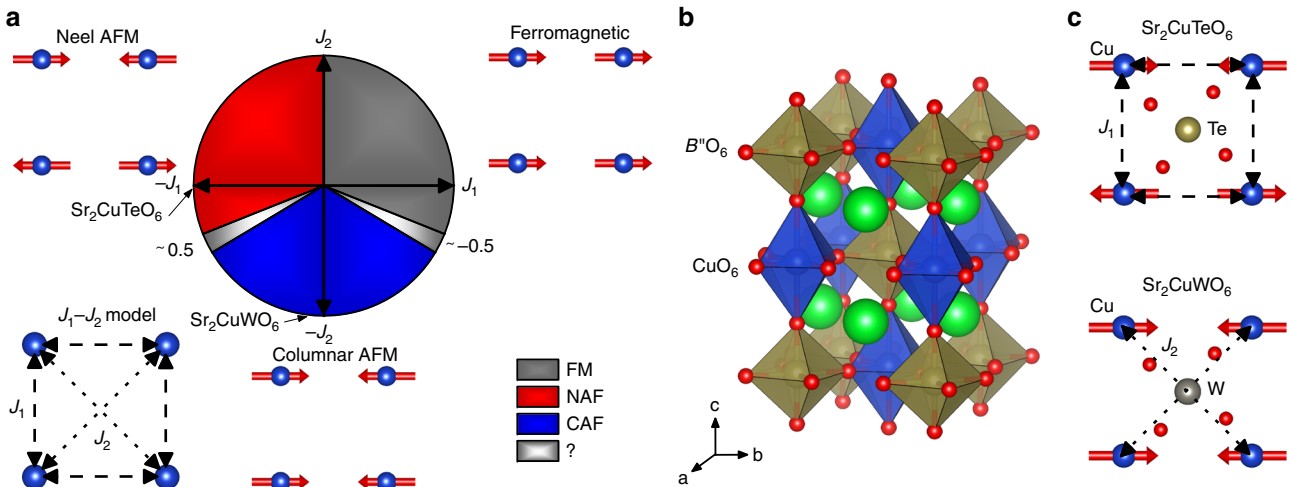

**Fig. 1** Spin-1/2 Heisenberg square-lattice model in $Sr_2CuTeO_6$ and $Sr_2CuWO_6$. **a** Phase diagram of the $J_1$–$J_2$ square-lattice model. $J_1$ is the nearest-neighbor interaction and $J_2$ the next-nearest-neighbor interaction. The classical ground states are ferromagnetic (FM), Néel antiferromagnetic (NAF) and columnar antiferromagnetic (CAF) ordering. The highly frustrated $J_2/J_1 \approx 0.5$ and $J_2/J_1 \approx -0.5$ regions are located at the NAF–CAF and CAF–FM boundaries, respectively. **b** The double perovskite structure of $Sr_2CuTeO_6$ and $Sr_2CuWO_6$. Sr, Cu, $B''$ (Te/W) and O are represented by green, blue, dark yellow and red spheres, respectively. The blue (dark yellow) octahedra represent $CuO_6$ ($B''O_6$). **c** The Néel antiferromagnetic structure of $Sr_2CuTeO_6$ and the columnar antiferromagnetic structure of $Sr_2CuWO_6$ with the view down the c-axis. The dominant antiferromagnetic interactions are shown

## Results

**Crystal structure.** Polycrystalline samples of $Sr_2Cu(Te_{0.5}W_{0.5})O_6$, $Sr_2CuTeO_6$ and $Sr_2CuWO_6$ with crystallite size in the micrometer range were synthesized via a conventional solid state reaction. Sample color ranged from light green to yellow, indicating that the materials are insulating. This was confirmed with a room-temperature four-probe electrical conductivity measurement. X-ray diffraction analysis found the samples to be of high quality with a trace $SrWO_4$ impurity in $Sr_2Cu(Te_{0.5}W_{0.5})O_6$ and $Sr_2CuWO_6$; the relatively stable $SrWO_4$ is a common impurity in $Sr_2CuWO_6$[15,21]. $Sr_2Cu(Te_{0.5}W_{0.5})O_6$ retains the $I4/m$ double perovskite structure of the parent phases with little difference in lattice parameters or bond distances; Rietveld refinement results are presented in Supplementary Fig. 1 and Supplementary Table 1. Cation order with respect to $B'$ ($Cu^{2+}$) and $B''$ ($Te^{6+}/W^{6+}$) sites is complete within experimental accuracy, but tellurium and tungsten are randomly distributed on the $B''$ site. This results in quenched disorder in the $J_1$ and $J_2$ interactions between the $Cu^{2+}$ ions.

**Magnetic properties.** Magnetic properties of $Sr_2Cu(Te_{0.5}W_{0.5})O_6$, $Sr_2CuTeO_6$ and $Sr_2CuWO_6$ are summarized in Table 1. DC magnetic susceptibilities as a function of temperature are presented in Fig. 2. The zero-field cooled (ZFC) and field cooled (FC) curves fully overlap in all samples, and therefore we present only the ZFC results. The magnetic susceptibilities of $Sr_2CuTeO_6$ and

**Table 1 Magnetic and thermodynamic properties of $Sr_2Cu(Te_{0.5}W_{0.5})O_6$, $Sr_2CuTeO_6$ and $Sr_2CuWO_6$**

|  | $Sr_2Cu(Te_{0.5}W_{0.5})O_6$ | $Sr_2CuTeO_6$ | $Sr_2CuWO_6$ |
|---|---|---|---|
| $T_{max}$ (K) | 52 | 74 | 86 |
| $\chi_{max}$ ($10^{-3}$ emu/mol) | 2.55 | 2.24 | 1.55 |
| $\mu_{eff}$ ($\mu_B$) | 1.87 | 1.87 | 1.90 |
| $\Theta_{cw}$ (K) | −71 | −80 | −165 |
| $T_N$ (K) | <0.019 | 29[24] | 24[17] |
| **k** | — | [½ ½ 0][24] | [0 ½ ½][23] |
| $f = |\Theta_{cw}|/T_N$ | >3700 | ≈3 | ≈7 |
| $\gamma$ (mJ/molK²) | 54.2 | 2.2 | 0.7 |
| $\beta_D$ (K) | 395 | 381 | 361 |

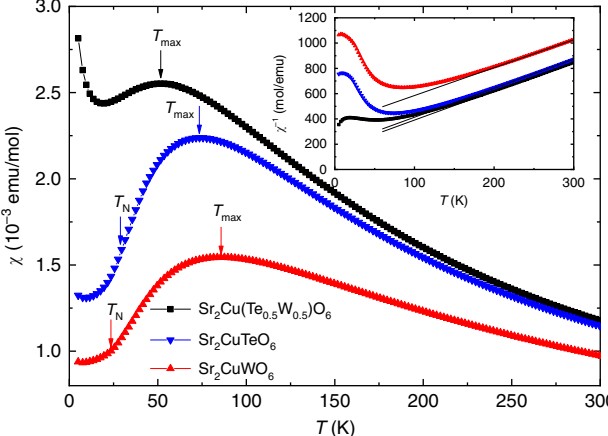

**Fig. 2** Magnetic susceptibility. DC magnetic susceptibility of $Sr_2Cu(Te_{0.5}W_{0.5})O_6$, $Sr_2CuTeO_6$ and $Sr_2CuWO_6$ measured in a 1 T field. Néel temperatures of $Sr_2CuTeO_6$ and $Sr_2CuWO_6$ are marked with $T_N$, whereas the position of the maximum in magnetic susceptibility is marked with $T_{max}$. Zero-field cooled and field cooled curves fully overlap and only the former is shown. Inset: Inverse magnetic susceptibility and fits to Curie–Weiss law

$Sr_2CuWO_6$ do not feature a cusp at $T_N$. Instead, in all three compounds we observe a broad maximum in the susceptibility, which is a common feature of two-dimensional magnets and QSL candidates[5]. This maximum can be characterized by two parameters: its position $T_{max}$ and height $\chi_{max}$. In the frustrated region of the square-lattice model near $J_2/J_1 \approx 0.5$ $T_{max}$ is predicted to be lower than in either the NAF ($J_2/J_1 \ll 0.5$) or CAF ($J_2/J_1 \gg 0.5$) regions[30,31]. Our magnetic data are consistent with this theoretical prediction: $T_{max}$ in $Sr_2Cu(Te_{0.5}W_{0.5})O_6$ shifts to a lower temperature than in $Sr_2CuTeO_6$ or $Sr_2CuWO_6$. This would place $Sr_2Cu(Te_{0.5}W_{0.5})O_6$ close to the highly frustrated region, although the structural disorder present in $Sr_2Cu(Te_{0.5}W_{0.5})O_6$ is not included in the theoretical model. In the related solid solution series $Sr_2Cu(W_{1-x}Mo_x)O_6$, where both end members have a dominating $J_2$ interaction and less frustration is expected, $T_{max}$ depends linearly on composition and never goes below those of the end members[15,17]. A Curie tail is observed in $Sr_2Cu(Te_{0.5}W_{0.5})O_6$ at low temperatures. This is likely to be from a paramagnetic impurity, which are known to be relatively common in the end phases[15–17,24].

Magnetic susceptibilities were fitted to the Curie–Weiss law $\chi = C/(T - \Theta_{cw})$, where $C$ is the Curie constant and $\Theta_{cw}$ is the Weiss constant. The inverse susceptibilities deviate from the linear Curie–Weiss behavior below relatively high temperatures of ≈ 200 K (inset in Fig. 2). For this reason, we performed the fitting in the temperature range 250–300 K. The Weiss constant $\Theta_{cw}$ gives an indication of the total strength of magnetic interactions in a material. For $Sr_2Cu(Te_{0.5}W_{0.5})O_6$ we obtain $\Theta_{cw} = -71$ K revealing mainly antiferromagnetic interactions similar in strength to those in $Sr_2CuTeO_6$ ($\Theta_{cw} = -80$ K). In contrast, the antiferromagnetic interactions in $Sr_2CuWO_6$ are significantly stronger with $\Theta_{cw} = -165$ K. Effective paramagnetic moments obtained from the Curie–Weiss fits are essentially the same for all samples and typical for $Cu^{2+}$ (Table 1). In DC susceptibility, the ZFC and FC curves were found to overlap for all samples, which indicates the lack of a spin glass transition. AC susceptibility of $Sr_2Cu(Te_{0.5}W_{0.5})O_6$ was measured (Supplementary Fig. 2) to support this conclusion. No frequency dependent peak was observed in the real part $\chi'$ (dispersion) of the AC susceptibility indicating that $Sr_2Cu(Te_{0.5}W_{0.5})O_6$ is not a spin glass. Moreover, the imaginary part $\chi''$ (absorption) remains practically zero.

**Specific heat.** Results of specific heat measurements of $Sr_2Cu(Te_{0.5}W_{0.5})O_6$, $Sr_2CuTeO_6$ and $Sr_2CuWO_6$ are shown in Fig. 3a. Similar to the magnetic susceptibility, $T_N$ cannot be simply determined from the specific heat of $Sr_2CuTeO_6$ or $Sr_2CuWO_6$ as no lambda anomalies are observed. Likewise, no lambda anomaly is seen for $Sr_2Cu(Te_{0.5}W_{0.5})O_6$ down to 2 K. Moreover, we do not observe a low-temperature maximum typical of spin-gapped systems such as valence bond solids[32,33] or the valence bond glass $Ba_2YMoO_6$[34]. The main difference between the compounds is that the reduced specific heat capacities of $Sr_2CuTeO_6$ and $Sr_2CuWO_6$ approach zero with decreasing temperature, as is expected for insulators, whereas the reduced specific heat of $Sr_2Cu(Te_{0.5}W_{0.5})O_6$ appears to remain finite.

At temperatures below ≈ 10 K, linear behavior is observed in a $C_p/T$ vs. $T^2$ plot (inset in Fig. 3a). Specific heat in the range 2–10 K was fitted using the function $C_p = \gamma T + \beta_D T^3$, where $\gamma$ is the $T$-linear electronic term and $\beta_D$ the Debye-like phononic term. The $T$-linear $\gamma$ terms obtained were 54.2(5), 2.2(2) and 0.7(4) mJ/molK² for $Sr_2Cu(Te_{0.5}W_{0.5})O_6$, $Sr_2CuTeO_6$ and $Sr_2CuWO_6$, respectively. $Sr_2Cu(Te_{0.5}W_{0.5})O_6$ has a notably large $\gamma$ term for an insulator with no free electrons. There are two main possibilities for a significant $\gamma$ term in an insulator. In a gapless

quantum spin liquid the $\gamma$ term arises from collective excitations of entangled spins[8,35]. On the other hand, the $\gamma$ term is also an archetypical feature of spin glasses. In spin glasses, $\gamma$ is associated with intrinsic spin disorder[36]. The $\gamma$ term can also develop above the spin freezing temperature $T_F$[37,38].

The specific heat of $Sr_2Cu(Te_{0.5}W_{0.5})O_6$ was also measured in a field of $\mu_0 H = 8$ T (Fig. 3a) and found to be nearly identical with the zero field measurements; a $\gamma$ term of 56.5(7) mJ/molK$^2$ was obtained. This lack of magnetic field dependency rules out a spin glass state[39], since both specific heat and the $\gamma$ term depend on the applied field in spin glasses[37]. The lack of field dependency indicates that the $\gamma$ term could be related to a fermionic density of states as is the case in quantum spin liquids[35]. The properties of the predicted spin-liquid state at $J_2/J_1 = 0.5$ are under debate. Recent theoretical work suggests either a gapless $Z_2$ QSL[40–42] or a topological e.g. gapped $Z_2$ QSL[43,44]. $Sr_2Cu(Te_{0.5}W_{0.5})O_6$ appears to have gapless excitations based on our specific heat measurements, but the existence of a small spin gap[14] cannot be ruled out.

For comparison, we measured the specific heat of similar double perovskite solid solutions (Supplementary Fig. 3). In non-magnetic $Sr_2Zn(Te_{0.5}W_{0.5})O_6$, the reduced specific heat approaches zero with decreasing temperature, yielding a small $\gamma$ term of 0.2(1) mJ/molK$^2$. This shows that the finite electronic term is related to the magnetism of $Cu^{2+}$ ions. We also measured the molybdenum analog $Sr_2Cu(Mo_{0.5}W_{0.5})O_6$, where both end members $Sr_2CuWO_6$ and $Sr_2CuMoO_6$ have a dominant $J_2$ interaction and thus less frustration is expected[17]. Here too the reduced specific heat approaches zero as

temperature is lowered and the electronic term is small, 0.7 (4) mJ/molK$^2$. We conclude that the significant $T$-linear term in the specific heat of $Sr_2Cu(Te_{0.5}W_{0.5})O_6$ is related to magnetic frustration and not solely to structural disorder.

An estimate of the magnetic specific heat of $Sr_2Cu(Te_{0.5}W_{0.5})O_6$ (Fig. 3b) was obtained by subtracting the specific heat of the closest non-magnetic analog $Sr_2Zn(Te_{0.5}W_{0.5})O_6$ (Fig. 3b). Unfortunately, the structure of $Sr_2ZnTe_{0.5}W_{0.5}O_6$ is slightly different, because it lacks the Jahn–Teller distortion present in $Sr_2Cu(Te_{0.5}W_{0.5})O_6$. This and weighting errors lead to some uncertainty in the removal of the phononic contribution, and thus we have scaled the lattice standard data to match the specific heat of both compounds in the high-temperature paramagnetic region[34]. The magnetic specific heat of $Sr_2Cu(Te_{0.5}W_{0.5})O_6$ increases with temperature up to $\approx 20$ K. In a gapless quantum spin liquid a linear increase in magnetic specific heat is expected at low temperatures[35,39]. In order to conclusively show this linear relationship in $Sr_2Cu(Te_{0.5}W_{0.5})O_6$, we present a log–log plot of magnetic specific heat as a function of temperature (Fig. 3c). A low-temperature fit to $\gamma T^\alpha$ yields $\alpha = 1.02(1)$, confirming the $T$-linear dependence of magnetic specific heat and $\gamma = 55(1)$ mJ/K consistent with the $C_p/T$ vs. $T^2$ fit. Magnetic entropy was integrated (Fig. 3d) and found to reach 2.51 J/molK at 90 K. This represents 44% of the expected $R\ln(2)$ for spin-1/2. Similar values have been reported for some other QSL candidates[39,45,46]. The large amount of spin entropy retained at low temperatures is a common feature of quantum spin liquids[39].

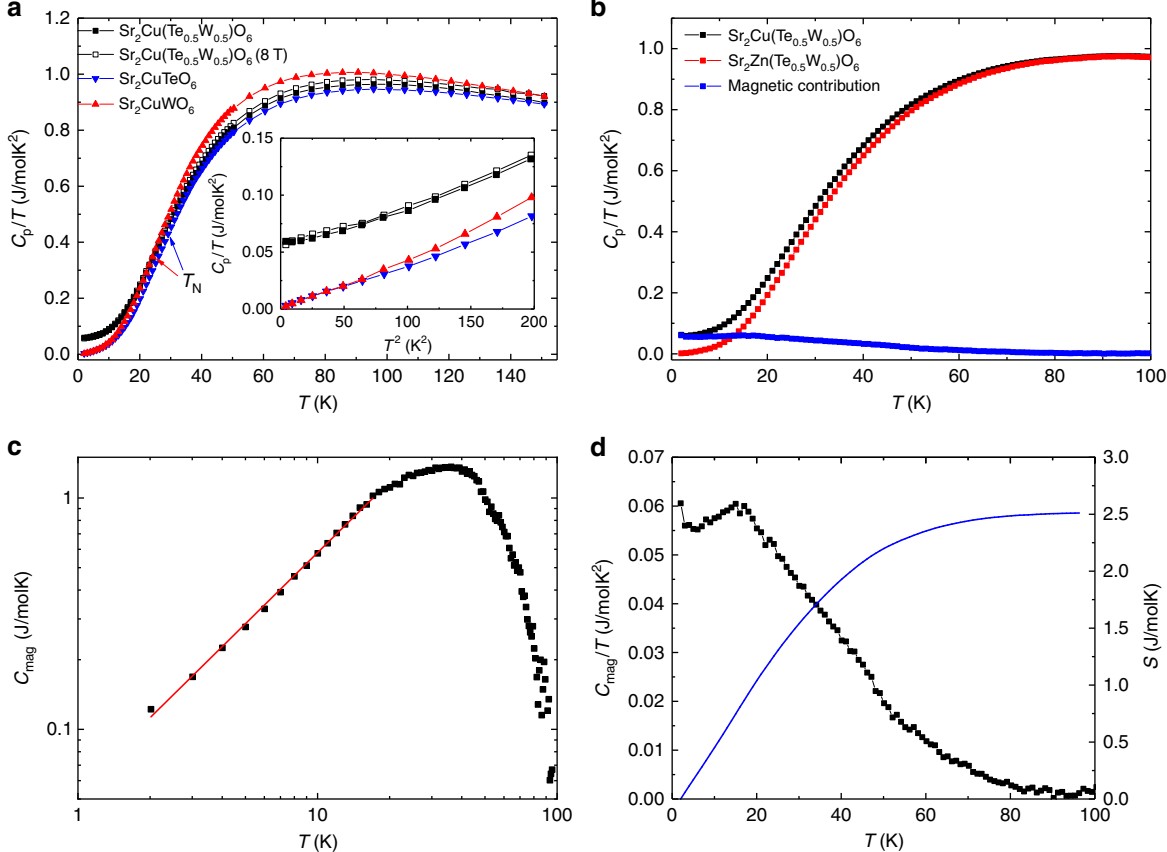

**Fig. 3** Specific heat measurements. **a** Specific heat of $Sr_2Cu(Te_{0.5}W_{0.5})O_6$, $Sr_2CuTeO_6$ and $Sr_2CuWO_6$. Inset: Low-temperature $C_p/T$ vs. $T^2$ plot. **b** Specific heat of $Sr_2Cu(Te_{0.5}W_{0.5})O_6$, lattice standard $Sr_2Zn(Te_{0.5}W_{0.5})O_6$ and the subtracted magnetic specific heat of $Sr_2Cu(Te_{0.5}W_{0.5})O_6$. **c** Log–log plot of magnetic specific heat of $Sr_2Cu(Te_{0.5}W_{0.5})O_6$. The red line is a fit to $\gamma T^\alpha$, which yields $\alpha = 1.02(1)$ confirming $T$-linear behavior. **d** Magnetic specific heat (black) and integrated entropy (blue) of $Sr_2Cu(Te_{0.5}W_{0.5})O_6$

**Muon spin rotation and relaxation**. No magnetic ordering or spin freezing could be observed for any of the samples in either magnetic susceptibility or specific heat measurements. For this reason, we measured muon spin rotation and relaxation (μSR) of $Sr_2Cu(Te_{0.5}W_{0.5})O_6$. μSR is a sensitive local probe of static and dynamic magnetism, and has been previously used to trace the onset of antiferromagnetic order in $Sr_2CuWO_6$[17]. We measured the μSR signal in zero field (ZF), weak transverse field (wTF) and longitudinal field (LF) modes down to 19 mK. Representative spectra taken in ZF mode at various temperatures are shown in Fig. 4a. We have plotted the time dependent polarization $G_z(t)$ of the initially 100% polarized muon spins measured via the asymmetry of decay positron count rates $a(t) = a(t=0)\cdot G_z(t)$. Even at 19 mK there is no indication of a spontaneous muon spin rotation signal (oscillations of asymmetry) expected for a magnetically ordered system. In contrast, clear oscillations are observed below $T_N$ in $Sr_2CuWO_6$[17]. Qualitatively, an increase in depolarization is observed with decreasing temperature. This means that the muon spins sense a distribution of local fields that is widened at lower temperatures. If these fields were of static origin, the distribution would correspond only to fields of a few $10^{-4}$ T, i.e., far less than expected for static local fields of $Cu^{2+}$ such as the $\approx 0.1$ T fields observed in $Sr_2CuWO_6$[17]. Moreover, the curves are clearly different from those of a magnetically frozen static spin system, where one expects a residual polarization of $G_z(t) \approx 1/3$ at long times. Thus, the muon spin relaxation appears to be mainly due to dynamic electronic spin fluctuations down to 19 mK.

In order to conclusively rule out magnetic ordering, we measured muon spin precession in a weak transverse field of 5 mT (Fig. 4b). The onset of magnetic ordering of $Cu^{2+}$ moments would cause a distinct loss of asymmetry in transverse field experiments due to randomly adding strong local magnetic fields to the weak applied transverse field. We find no loss of initial asymmetry $a(t=0)$ in the wTF experiments down to 19 mK. Damping increases upon decreasing temperature and tends to saturate below 0.5 K in parallel to the behavior observed in ZF (see below).

A quantitative description of the ZF spectra was obtained using a power law function $G_z(t) = \exp(-(\lambda t)^\beta)$ as the depolarization function, where $\lambda$ is the relaxation rate and $\beta$ is a power factor. This approach has also been used in the analysis of the Kagomé-lattice spin-liquid system $SrCr_8Ga_4O_{19}$[47]. The fits shown in Fig. 4a were obtained using this phenomenological approach, which should be applicable when the field fluctuation rates are larger than the damping rates by static fields. The spectra above 2 K reveal close to exponential damping with $\beta = 1$, indicating fast fluctuating local fields, whereas $\beta$ increases from 1 to $\approx 1.8$ (Fig. 4c) at lower temperatures as seen from the more Gaussian-like appearance of the spectra. A Gaussian shaped ($\beta = 2$) depolarization function is typical for a static Gaussian local field distribution caused by disordered magnetic moments in the nearest neighbors of muons. The observed increase of $\beta$ shows that the fluctuation rates of local fields are slowing down at lower temperatures. The relaxation rate $\lambda$ increases at lower temperatures (Fig. 4c), which also corresponds to a decrease in the local

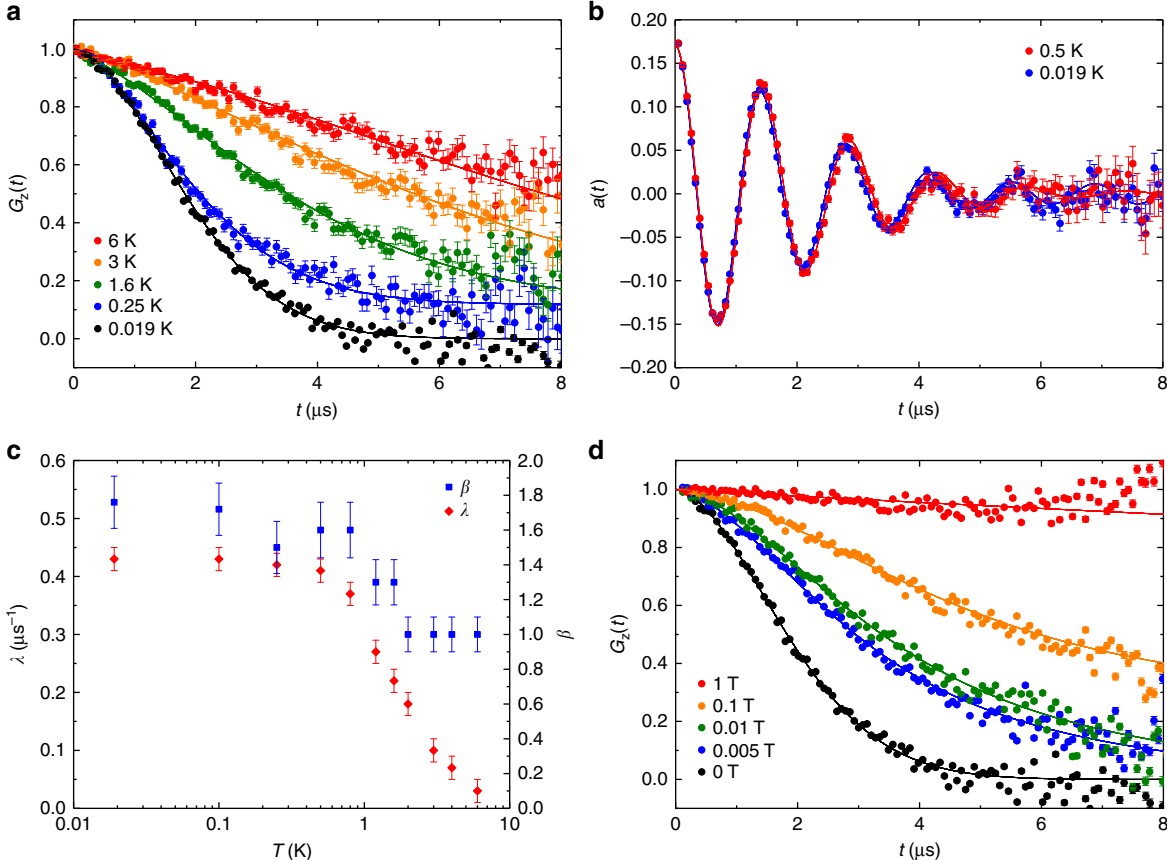

**Fig. 4** μSR measurements of $Sr_2Cu(Te_{0.5}W_{0.5})O_6$. **a** Zero-field muon spin relaxation function $G_z(t)$ measured at different temperatures. **b** Muon spin rotation spectra $a(t)$ measured with a 5 mT transverse-field at 19 mK and 0.5 K. **c** Muon spin relaxation rate $\lambda$ and power $\beta$ as a function of temperature from fits of the zero-field data using $\exp(-(\lambda t)^\beta)$ as the depolarization function. **d** Longitudinal-field muon spin relaxation function $G_z(t)$ at 19 mK measured with different applied longitudinal fields. The error bars in **a**, **b** and **d** represent 1 s.d. and in (**c**) the maximum possible variation due to correlation of parameters

field fluctuation rates sensed by the muon spins. At very low temperatures, between 0.5 K and 19 mK, $\lambda$ levels are off. Notably, this plateau in $\lambda$ is expected behavior in quantum spin liquid candidates[9,48,49].

Further proof of the persistence of relatively fast electronic spin dynamics even at 19 mK is provided by LF measurements (Fig. 4d). Longitudinal fields can suppress muon spin depolarization caused by weaker local static fields. Weak randomly oriented static fields of the order of mT or less are typically due to nuclear dipole moments (in the present case mainly from $^{63}$Cu and $^{65}$Cu nuclei). Depolarization by fast fluctuating local fields from atomic spins, however, may only be affected by much larger applied fields. The LF spectra in Fig. 4d show that a 5 mT LF is only enough for a partial suppression of the depolarization of muon spins. This means that these muon spins are in positions where weak static random local fields from nearby nuclear spins are acting. The depolarization of the majority of muon spins, however, is only gradually reduced by much higher fields. Depolarization becomes nearly completely suppressed by an applied LF of 1 T. This is typical for depolarization caused by fast fluctuating local fields. Whether the observation of two (or several) muon ensembles is related to local inhomogeneities in the solid solution or different muon sites in the lattice is an open question. This situation also precludes a detailed quantitative analysis of the muon spin relaxation under applied field using, e.g., Keren's function involving a single fluctuation rate and a single static damping[50]. From the variation of damping as a function of applied field, we may, however, estimate the field fluctuation rates at 19 mK to be on the order of 80–100 MHz.

## Discussion

We have successfully used $d^{10}$–$d^0$ cation mixing to tune the magnetic ground state of the spin-1/2 square-lattice antiferromagnet $Sr_2Cu(Te_{0.5}W_{0.5})O_6$, which retains the double perovskite structure of its parent phases $Sr_2CuTeO_6$ and $Sr_2CuWO_6$. The broad maximum in the magnetic susceptibility shifts to a lower temperature in the solid solution indicating increased frustration. The specific heat of $Sr_2Cu(Te_{0.5}W_{0.5})O_6$ has a significant $T$-linear term of $\gamma = 54.2(5)$ mJ/molK$^2$ despite the phase being an insulator. A complete lack of static magnetism down to 19 mK is revealed by muon spin rotation and relaxation measurements. This corresponds to a frustration factor $f = |\Theta_{cw}|/T_N$ of over 3700. Moreover, the muon spin relaxation rate has a clear plateau at low temperatures. Our experimental results therefore indicate a spin-liquid-like ground state in $Sr_2Cu(Te_{0.5}W_{0.5})O_6$. This is the first observation of such a ground state in a square-lattice compound.

The origin of the spin-liquid-like state in $Sr_2Cu(Te_{0.5}W_{0.5})O_6$ remains unclear. The $J_1$–$J_2$ model provides an appealingly simple explanation for our experimental observations, as they are consistent with the $Z_2$ gapless quantum spin liquid state predicted for $J_2/J_1 = 0.5$. However, neither the average values of magnetic interactions $J_1$ and $J_2$ nor the applicability of the model itself due to disorder is known at this time. An alternative origin could be in the combination of disorder and magnetic frustration in the material. Disorder and magnetic frustration are inherently linked in $Sr_2Cu(Te_{0.5}W_{0.5})O_6$: each elementary $Cu^{2+}$ square, or plaquette, has randomly either a $Te^{6+}$ ($d^{10}$) or a $W^{6+}$ ($d^0$) cation in its center promoting a dominant $J_1$ or $J_2$ interaction, respectively. As a consequence, $Sr_2Cu(Te_{0.5}W_{0.5})O_6$ has very significant quenched disorder in the magnetic interactions between spin-1/2 sites. Disorder has been shown to induce a gapless spin-liquid state in a spin-1/2 triangular compound[51]. Recently, Kawamura and coworkers[52–55] proposed that a disorder-induced gapless quantum spin liquid state, a so-called random-singlet state, forms on spin-1/2 triangular, Kagomé and honeycomb lattices when

there is a high amount of randomness in the magnetic interactions in addition to frustration. They suggested that the random-singlet state could be found in many frustrated compounds with disorder in a wide parameter range without needing an exact match of magnetic interactions[55]. $Sr_2Cu(Te_{0.5}W_{0.5})O_6$, with its unique combination of significant disorder and magnetic frustration, could be a realization of this random-singlet state on the square lattice.

The $d^{10}$–$d^0$ cation mixing approach utilized here could be used to tune the ground state in other quantum materials such as those close to a quantum critical point. In this work, we have utilized the similar size of $d^{10}$ $Te^{6+}$ and $d^0$ $W^{6+}$ cations. We can identify the additional $d^{10}/d^0$ cation pairs of $Zn^{2+}/Mg^{2+}$, $Cd^{2+}/Ca^{2+}$, $In^{3+}/Sc^{3+}$ and $Sb^{5+}/Nb^{5+}$ based on ionic radii[22] and previous work by Marjerrison and coworkers[29].

## Methods

**Sample synthesis**. Polycrystalline powders of $Sr_2Cu(Te_{0.5}W_{0.5})O_6$, $Sr_2CuTeO_6$ and $Sr_2CuWO_6$ were prepared by a solid-state reaction method. Stoichiometric amounts of $SrCO_3$, CuO, $TeO_2$ and $WO_3$ (all 99.995% or greater purity, Alfa Aesar) were thoroughly ground in an agate mortar with ethanol. The samples were calcined at 900 °C in air for 12 h, thoroughly ground, pressed into pellets, and fired at 1050 °C in air for 72 h with intermittent grindings.

**Xray diffraction**. Crystal structure and phase purity were evaluated by powder xray diffraction. Diffraction patterns were collected at room temperature with a Panalytical X'Pert Pro MPD instrument (Cu $K_{\alpha 1}$ radiation). Refinement of the crystal structure was carried out using FULLPROF[56] software. Crystallite size was evaluated using line broadening analysis[57]. The crystal structures were visualized using VESTA[58].

**Magnetic characterization**. Magnetic properties were investigated using a Quantum Design MPMS XL SQUID magnetometer. 100 mg of the samples were enclosed in gelatin capsules and placed in plastic straws. DC magnetic susceptibility was measured in zero-field cooled and field cooled modes between 5 K and 300 K in an applied field of $\mu_0 H = 1$ T. AC susceptibility measurements were performed using a Quantum Design PPMS-9 on a 41.5 mg pressed pellet sample of $Sr_2Cu(Te_{0.5}W_{0.5})O_6$. The sample was cooled in zero magnetic field and then measured at a range of frequencies between 215 and 9899 Hz, with an AC amplitude of 1 mT and a DC field of 0.5 T, as a function of temperature.

**Specific heat**. Specific heat capacity measurements were done with a Quantum Design PPMS between 2 K and 150 K using a thermal relaxation method. The samples were pieces of pellets with masses of 8–10 mg. Specific heat was measured in zero field and for $Sr_2Cu(Te_{0.5}W_{0.5})O_6$ also in $\mu_0 H = 8$ T. Magnetic specific heat of $Sr_2Cu(Te_{0.5}W_{0.5})O_6$ was obtained by subtracting the specific heat of $Sr_2Zn(Te_{0.5}W_{0.5})O_6$ after scaling the latter to fit the high-temperature paramagnetic part.

**Muon spin rotation and relaxation (µSR)**. The experiments were performed on a polycrystalline powder sample of $Sr_2Cu(Te_{0.5}W_{0.5})O_6$ using the 100% spin polarized surface muon beam at the Dolly ($^3$He cryostat down to 0.25 K) and LTF ($^3$He/$^4$He dilution refrigerator down to 19 mK) facilities of the Swiss Muon Source at the Paul Scherrer Institut, Switzerland. The measurements were made in zero-field, longitudinal field (along initial muon spin) and transverse field (perpendicular to initial muon spin) modes.

**Data availability**. All the data supporting the conclusions of this article are available from the authors upon reasonable request.

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

## Acknowledgements

Dr. J-C. Orain is thanked for technical assistance with the μSR measurements at Dolly. Dr. E.M. Bittar is thanked for the use of laboratory facilities. We thank Dr. G. Stenning for help with AC susceptibility measurements in the Materials Characterization Laboratory at the ISIS Neutron and Muon Source. O.M. is grateful for an Aalto CHEM funded doctoral student position. S.V. is thankful for the support of the Brazilian funding agencies CNPq (grants no. 150503/2016-4 and 152331/2016-6) and FAPERJ (grant no. 202842/2016). F.J.L, E.S. and E.B-S. are grateful for financial support by a joint DFG-FAPERJ project DFG Li- 244/12. In addition, E.B-S. acknowledges support from FAPERJ

through several grants, including Emeritus Professor fellow and CNPq for BPA and corresponding grants.

## Author contributions

O.M., S.V. and M.K. conceived and planned the study. The samples were synthesized by O.M. and S.V. Structural characterization and analysis was performed by O.M. and S.V. Magnetic properties were measured by O.M. and H.C.W. Specific heat was measured by S.V. and E.B-S. Magnetic and specific heat data were analyzed and interpreted by O.M., S.V., H.C.W., I.T., E.B-S. and M.K. Muon spin rotation and relaxation was measured by O.M., E.S., K.P.S., C.B. and F.J.L. with the main analysis by F.J.L. and K.P.S. O.M., S.V. and F.J.L. wrote the manuscript with contributions from all authors.

## Additional information

**Competing interests:** The authors declare no competing interests.

