## [Peer Review File · Nature Communications]

Reviewers' comments:

Reviewer #1 (Remarks to the Author):

Mustonen and coworkers report magnetization, specific heat and μ SR measurements on polycrystalline samples of $\text{Sr}_2\text{Cu}(\text{Te}_{1-x}\text{W}_x)\text{O}_6$ for $x=1$, 0.5 and 0. The $x=1$ and $x=0$ compounds are $S=1/2$ systems on a square lattice with competing nearest neighbour J_1 and next nearest neighbour J_2 interactions. Previous neutron scattering measurements showed that while the $x=0$ is characterized by a Néel antiferromagnetic order the $x=1$ shows a columnar order. In between those two types of order a spin liquid phase is expected. The authors have concentrated on the $x=0.5$ compound which is expected to be a mixture of the local configuration expected for $x=0$ with a low J_2/J_1 ratio and of the $x=1$ one with a $J_2/J_1 \gg 1$.

Magnetization measurements evidence an increase of frustration for the $x=0.5$ sample, while μ SR measurements clearly show that the $x=0.5$ sample, unlike the other two, does not show any magnetic long-range order down to 0.25 K. Specific heat measurements reveal a continuum of low-energy excitations which is consistent with a gapless spin liquid phase. While the presence of a spin liquid phase does remain a possibility I don't think that the μ SR measurements shown here unambiguously support that scenario. As also the authors point out, in the presence of a degenerate spin liquid ground-state one would expect a plateau in the μ SR relaxation. Here the "plateau" is evidenced by just two relaxation points (see Fig.4c). The data in Fig.4c show a slowing down of the fluctuations on cooling which could be present also if there is a magnetic order below 0.25 K.

How the disordered configuration of the $x=0.5$ sample maps onto an effective J_1 - J_2 hamiltonian, introduced in the first part, and which is the connection with the phase diagram shown in Fig.1 is not at all straightforward. In particular, would the spin-liquid phase be the one expected for the J_1 - J_2 model for J_2/J_1 around 0.5? The authors point out that the disorder may have a relevant role, as predicted for geometrically frustrated magnets with random bonds. However, it is not trivial to transfer those predictions to the J_1 - J_2 model on a square lattice with a mixture of large and low J_2/J_1 local configuration.

Moreover I have some further remarks that the authors should consider:

a) Have the authors tried to derive the characteristic time scale of the fluctuations by fitting the data in Fig.4d with a dynamical Kubo-Toyabe function or even with Keren function? This would allow to better understand what is the observed slowing down and what is the temperature dependence of the characteristic time scale. Notice that in the J_1 - J_2 model nematic-like low frequency fluctuations can be present (see Phys.Rev.Lett. 88, 047601 (2002)) and give rise to a similar trend in the relaxation.

b) The authors point out that the total entropy expected for a $S=1/2$ system is not fully recovered at high temperature. This suggests that there is a residual zero-temperature entropy associated with the degeneracy of the ground-state. Would this zero-temperature entropy be consistent with what is expected?

c) In these compounds the Curie-Weiss temperature should be given by J_1+J_2 . It is not clear to me why mixing configurations with a $|\Theta_{\text{CW}}| = 80$ K and 165 K would lead to a lower value for the $x=0.5$ phase. Can the authors clarify this point?

Overall, the manuscript could potentially be interesting, however in the present form there are not enough experimental data supporting their main conclusions nor there is a full understanding of why a spin-liquid phase is arising. Hence, I do not find this manuscript sound enough, at least in the present form, to warrant publication in Nature Communications.

Reviewer #2 (Remarks to the Author):

I have had the opportunity to review the manuscript, "Suppression of magnetic order in a spin-1/2 square-lattice antiferromagnet perovskite by d10-d0 cation ordering," by Mustonen et al. The authors have detailed the synthesis and characterization of a new spin liquid candidate $\text{Sr}_2\text{Cu}(\text{Te}_{0.5}\text{W}_{0.5})\text{O}_6$. This is a very good piece of work that should be seriously considered by the editors for publication in Nature Communications. The paper details a comprehensive analysis of the low temperature properties of a new $S=1/2$ quantum spin liquid. Square-lattice antiferromagnets have direct links to other quantum materials such as the high- T_c cuprates, and a detailed understanding of their behaviour is important for advancing the field.

However, I did have some suggestions for improving the work before publication. Some of these issues are minor, but they are important towards obtaining the best possible physical parameters to characterize this material:

(1) In figure 3, the specific heat is plotted in panel (a). It would be beneficial to also plot the lattice standard to see the subtraction properly for figure (b). I realize that this is in the Supplementary Material, but it should be shown in the main paper to guide the reader.

(2) It appears that the lattice standard has data points ABOVE the magnetic material at high T. This indicates that it is not a good lattice standard, or that there is a systematic error to the measurements. This needs to be addressed before publication. I do not think that it will change the curvature of the specific heat much at low T (so there will still be a linear response). However, it will change the integrated entropy release.

(3) In figure 3(b), the low T C_{mag}/T data is rather noisy. Is this truly linear? A fit would be preferred, with a log-log plot being the standard way to show linearity (with an error analysis of the slope to obtain the correct exponent). There is also a typo in the caption ("Subtracted").

(4) It is curious that the Curie-Weiss temperature of $\text{Sr}_2\text{Cu}(\text{Te}_{0.5}\text{W}_{0.5})\text{O}_6$ does not change very much from $\text{Sr}_2\text{CuTeO}_6$ (both are close to -80 K). This must be due to competing exchange interactions. Based on what is known about J_1/J_2 from inelastic neutron scattering experiments, could the authors model the Curie-Weiss temperature through an explicit calculation for the end members, and then attempt to apply this to the disordered case through extrapolation? This would be useful towards extracting where $\text{Sr}_2\text{Cu}(\text{Te}_{0.5}\text{W}_{0.5})\text{O}_6$ appears on the phase diagram.

Reviewer #3 (Remarks to the Author):

This work will be of interest to many in the condensed matter community. I find the results convincing and I would expect the paper to be highly cited.

Figure 2 shows the magnetic susceptibility. The authors say they measured under both FC and ZFC conditions but they do not say which data they have plotted. They are at pains to emphasise that various aspects of their results are inconsistent with the formation of a spin glass at the 1:1 composition and overlapping ZFC and FC curves would provide further evidence.

There is one fundamental point that I don't fully understand. I can see that the switch from a full to an empty d shell will modify the strength of the $J_2 \text{Cu} - \text{O} - \text{B}' - \text{O} - \text{Cu}$ interaction and it will therefore change the J_1/J_2 ratio. However, the authors don't explain why Te vs W should cause a

6-fold change in the strength of the J_1 Cu - O - O - Cu interaction, which it apparently does. Any ideas?

RE: NCOMMS-17-26462-T

Summary of Changes

- We have changed the title to “Spin-liquid-like state in a spin-1/2 square-lattice antiferromagnet perovskite induced by d^{10} - d^0 cation mixing” to better describe the novelty of our work. Due to new experiments we have also added one author: Mr. Christopher Baines.
- We have performed new μ SR experiments down to 19 mK to provide more convincing evidence for a spin-liquid-like state. We have revised Fig. 4 and made changes to the μ SR section of the manuscript.
- We have revised Fig 3 (specific heat) including a log-log plot of the magnetic specific heat
- We have added comparisons to theoretical predictions on the square-lattice and to other spin-liquid candidates at many points in the manuscript.
- The Discussion section has been rewritten to emphasize, that the origin of the spin-liquid-like state is somewhat unclear but could be the random-singlet state proposed by Kawamura.

Answers to Reviewers' Comments

Reviewer #1

Mustonen and coworkers report magnetization, specific heat and μ SR measurements on polycrystalline samples of $\text{Sr}_2\text{Cu}(\text{Te}_{1-x}\text{W}_x)\text{O}_6$ for $x = 1, 0.5$ and 0 . The $x = 1$ and $x = 0$ compounds are $S = 1/2$ systems on a square lattice with competing nearest neighbour J_1 and next nearest neighbour J_2 interactions. Previous neutron scattering measurements showed that while the $x = 0$ is characterized by a Néel antiferromagnetic order the $x=1$ shows a columnar order. In between those two types of order a spin liquid phase is expected. The authors have concentrated on the $x = 0.5$ compound which is expected to be a mixture of the local configuration expected for $x = 0$ with a low J_2/J_1 ratio and of the $x = 1$ one with a $J_2/J_1 \gg 1$.

(1) Magnetization measurements evidence an increase of frustration for the $x = 0.5$ sample, while μ SR measurements clearly show that the $x = 0.5$ sample, unlike the other two, does not show any magnetic long-range order down to 0.25 K. Specific heat measurements reveal a continuum of low-energy excitations which is consistent with a gapless spin liquid phase. While the presence of a spin liquid phase does remain a possibility I don't think that the μ SR measurements shown here unambiguously support that scenario. As also the authors point out, in the presence of a degenerate spin liquid ground-state one would expect a plateau in the μ SR relaxation. Here the “plateau” is evidenced by just two relaxation points (see Fig. 4c). The data in Fig. 4c show a slowing down of the fluctuations on cooling which could be present also if there is a magnetic order below

0.25

K.

Answer: We completely agree with the referee that the expected plateau in relaxation rate was not unambiguously shown in the original manuscript. We received additional beamtime at the Low-Temperature Facility of the Swiss Muon Source allowing us to carry out further experiments on the $x = 0.5$ sample. Our new μ SR data, measured down to 19 mK (250 mK in the original manuscript), convincingly show that the relaxation rate exhibits a plateau between 500 mK and 19 mK. Accordingly, we have revised Figure 4, Table I and the main text.

(2) How the disordered configuration of the $x=0.5$ sample maps onto an effective J1-J2 hamiltonian, introduced in the first part, and which is the connection with the phase diagram shown in Fig.1 is not at all straightforward. In particular, would the spin-liquid phase be the one expected for the J1-J2 model for $J2/J1$ around 0.5 ? The authors point out that the disorder may have a relevant role, as predicted for geometrically frustrated magnets with random bonds. However, it is not trivial to transfer those predictions to the J1-J2 model on a square lattice with a mixture of large and low $J2/J1$ local configuration.

Answer: We agree with the referee that the origin of the spin-liquid-like state is not clear. We have rewritten the Discussion section to emphasize this point. As pointed out by the referee, the spin-liquid-like state in $x = 0.5$ appears to be gapless based on the specific heat data. Predictions for the $J2/J1 = 0.5$ region include both a gapless Z2 QSL and topological spin-liquids such as the gapped Z2 QSL. We have added references to theoretical papers and a brief comparison to the text.

Prof. Kawamura has made a similar prediction of a random-singlet state also on the square-lattice, but this work has not been published yet. A meeting abstract describing the result is available online:

<https://aquarius.mp.es.osaka-u.ac.jp/spintronics/hp/kanamori-memorial/abstract-session-quantumspin/Q4-Kawamura.pdf>

In this abstract prof. Kawamura argues that a gapless random singlet state forms on the square-lattice in a wide parameter range in the presence of disorder. This is an expansion of predictions made on the honeycomb J1-J2 model [Uematsu and Kawamura JPSJ 86 044704]. In a future work we plan to further investigate this possibility by looking at the full $\text{Sr}_2\text{Cu}(\text{Te}_{1-x}\text{W}_x)\text{O}_6$ solid solution series. If $\text{Sr}_2\text{Cu}(\text{Te}_{0.5}\text{W}_{0.5})\text{O}_6$ is a manifestation of this random-singlet state, one would expect this state to exist at a relatively wide composition range. If, on the other hand, the origin of the spin-liquid-like state in $\text{Sr}_2\text{Cu}(\text{Te}_{0.5}\text{W}_{0.5})\text{O}_6$ is in the exact match of J1 and J2, the spin-liquid-like state should only exist in a narrow composition range.

We have clarified in the Discussion section that the random singlet state is an alternative explanation for the origin of the spin-liquid-like state. We have added a sentence describing that an exact match of J1 and J2 are not needed in this model (if there is disorder). We have also added a reference to another spin-1/2 system where disorder was shown to induce a gapless spin liquid state.

In most theoretical investigations of disorder in magnetic materials disorder is treated as random bonds, i.e. the interactions have a distribution with an average and a variance. The referee rightfully points out that this differs from the expected local disorder in magnetic interactions caused by the d^{10} - d^0 cation mixing in the $x = 0.5$ sample. Nevertheless, we feel that these random bond models provide a reasonable starting point for understanding disorder in these materials. To the best knowledge of the authors the specific type of structural disorder found in $\text{Sr}_2\text{Cu}(\text{Te}_{0.5}\text{W}_{0.5})\text{O}_6$

combined with magnetic frustration has never been investigated in a $S = 1/2$ system before. As experimentalists, we hope that this work will inspire further theoretical investigations of frustrated square-lattice antiferromagnets with disorder.

(3) Moreover, I have some further remarks that the authors should consider:

(a) Have the authors tried to derive the characteristic time scale of the fluctuations by fitting the data in Fig. 4d with a dynamical Kubo-Toyabe function or even with Keren function? This would allow to better understand what is the observed slowing down and what is the temperature dependence of the characteristic time scale. Notice that in the J1-J2 model nematic-like low frequency fluctuations can be present (see Phys. Rev. Lett. 88, 047601 (2002)) and give rise to a similar trend in the relaxation.

Answer: Fits using a dynamic KT function or Keren's function were attempted, yet with limited success due to the apparent distribution of static dampings and correlation times. We have added a sentence to clarify this. Irrespective of the quantitative analysis of spectral shape we are able to give an estimate of the fluctuation rates with about 80-100 MHz (added in text). A detailed analysis will be presented separately. Concerning the paper by Carretta et al: The decoupling behavior is similar, but in fact we need much larger fields for decoupling corresponding to the much higher fluctuation rates.

(b) The authors point out that the total entropy expected for a $S = 1/2$ system is not fully recovered at high temperature. This suggests that there is a residual zero-temperature entropy associated with the degeneracy of the ground-state. Would this zero-temperature entropy be consistent with what is expected?

Answer: The amount of unrecovered entropy has some uncertainty, see Referee #2 remark (2). The amount of unrecovered entropy in spin-liquid candidates varies and we have added a brief comparison.

(c) In these compounds the Curie-Weiss temperature should be given by $J1+J2$. It is not clear to me why mixing configurations with a $|\Theta_{CW}| = 80$ K and 165 K would lead to a lower value for the $x=0.5$ phase. Can the authors clarify this point?

Answer: As we answer to Referee #2, we do not know the average J1 and J2 interactions. One possible explanation would be in minute changes in crystal structure, mainly in terms of rotation of the octahedra. It has previously been proposed, that the Weiss temperature is related to the Cu – O – Te/W angle in these materials [Iwanaga et al. JSSC 147 291 (1999)] mainly to explain why barium equivalents have stronger interactions. This angle is, in fact, slightly smaller in $x = 0.5$ than in $x = 0$.

Overall, the manuscript could potentially be interesting, however in the present form there are not enough experimental data supporting their main conclusions nor there is a full understanding of why a spin-liquid phase is arising. Hence, I do not find this manuscript sound enough, at least in the present form, to warrant publication in Nature Communications.

Reviewer #2

I have had the opportunity to review the manuscript, "Suppression of magnetic order in a spin-1/2 square-lattice antiferromagnet perovskite by d^{10} - d^0 cation ordering," by Mustonen et al. The authors have detailed the synthesis and characterization of a new spin liquid candidate $\text{Sr}_2\text{Cu}(\text{Te}_{0.5}\text{W}_{0.5})\text{O}_6$. This is a very good piece of work that should be seriously considered by the editors for publication in Nature Communications. The paper details a comprehensive analysis of the low temperature properties of a new $S=1/2$ quantum spin liquid. Square-lattice antiferromagnets have direct links to other quantum materials such as the high- T_c cuprates, and a detailed understanding of their behaviour is important for advancing the field.

However, I did have some suggestions for improving the work before publication. Some of these issues are minor, but they are important towards obtaining the best possible physical parameters to characterize this material:

(1) In figure 3, the specific heat is plotted in panel (a). It would be beneficial to also plot the lattice standard to see the subtraction properly for figure (b). I realize that this is in the Supplementary Material, but it should be shown in the main paper to guide the reader.

Answer: We have taken this suggestion and point (3) into account by making the following changes to Fig. 3. A new panel b shows the specific heat of $\text{Sr}_2\text{Cu}(\text{Te}_{0.5}\text{W}_{0.5})\text{O}_6$, $\text{Sr}_2\text{Zn}(\text{Te}_{0.5}\text{W}_{0.5})\text{O}_6$ and the subtracted magnetic specific heat. Panel c) shows the log-log plot confirming T-linear behavior and panel d) shows a closer view of the magnetic specific heat and integrated entropy.

(2) It appears that the lattice standard has data points ABOVE the magnetic material at high T. This indicates that it is not a good lattice standard, or that there is a systematic error to the measurements. This needs to be addressed before publication. I do not think that it will change the curvature of the specific heat much at low T (so there will still be a linear response). However, it will change the integrated entropy release.

Answer: We have re-measured the specific heat of $\text{Sr}_2\text{Cu}(\text{Te}_{0.5}\text{W}_{0.5})\text{O}_6$ and $\text{Sr}_2\text{Zn}(\text{Te}_{0.5}\text{W}_{0.5})\text{O}_6$ with smaller temperature steps to further investigate the magnetic specific heat. The results of the new measurements are very similar to our previous measurements but did lead to a small change in the entropy as predicted by the referee.

We agree with the referee that the Zn compound might not be an ideal lattice standard. As we have explained in the text, having a d^{10} shell it lacks the Jahn-Teller distortion found in $\text{Sr}_2\text{Cu}(\text{Te}_{0.5}\text{W}_{0.5})\text{O}_6$ and thus has a slightly different structure. A compound with JT distortion, on the other hand, would be magnetic itself. We have clarified in the text that due to this issue and weighting error we have matched the high-temperature specific heat of these two compounds in the paramagnetic region.

(3) In figure 3(b), the low T C_{mag}/T data is rather noisy. Is this truly linear? A fit would be preferred, with a log-log plot being the standard way to show linearity (with an error analysis of the slope to obtain the correct exponent). There is also a typo in the caption ("Subtracted").

Answer: We have re-measured the data as described above and added a log-log plot to Fig. 3c as suggested. We obtained an exponent of 1.02(1) confirming linearity.

(4) It is curious that the Curie-Weiss temperature of $\text{Sr}_2\text{Cu}(\text{Te}_{0.5}\text{W}_{0.5})\text{O}_6$ does not change very much from $\text{Sr}_2\text{CuTeO}_6$ (both are close to -80 K). This must be due to competing exchange interactions. Based on what is known about J_1/J_2 from inelastic neutron scattering experiments, could the authors model the Curie-Weiss temperature through an explicit calculation for the end members, and then attempt to apply this to the disordered case through extrapolation? This would be useful towards extracting where $\text{Sr}_2\text{Cu}(\text{Te}_{0.5}\text{W}_{0.5})\text{O}_6$ appears on the phase diagram.

Answer: As pointed out by Referee #1, it is not fully known if $\text{Sr}_2\text{Cu}(\text{Te}_{0.5}\text{W}_{0.5})\text{O}_6$ is well described by the J_1 - J_2 model due to the disorder in magnetic interactions. We completely agree that the average values of J_1 and J_2 would be of great interest. If one makes the crudest of approximations and simply interpolates J_1 and J_2 from the end phases, one gets $J_1 = -4.19$ and $J_2 = -4.89$ placing $\text{Sr}_2\text{Cu}(\text{Te}_{0.5}\text{W}_{0.5})\text{O}_6$ in the columnar region of the phase diagram. The same method would yield $x \sim 0.3$, i.e. $\text{Sr}_2\text{Cu}(\text{Te}_{0.7}\text{W}_{0.3})\text{O}_6$ for the composition where $J_2/J_1 = 0.5$.

We have attempted to obtain the average J_1 and J_2 by fitting the magnetic susceptibility using a high-temperature expansion method [Rosner et al. PRB 67 014416 (2003)]. This method works reasonably well for the end members. Unfortunately for $\text{Sr}_2\text{Cu}(\text{Te}_{0.5}\text{W}_{0.5})\text{O}_6$ the fitting was not particularly stable with regards to the chosen temperature range. Crucially, this method always results in two equivalent solutions (pairs of J_1 and J_2): in our case one with stronger J_1 and one with stronger J_2 . For these reasons, we were unable to obtain a reasonable estimate of the average J_1 and J_2 . We plan to investigate this in a future work.

Reviewer #3

This work will be of interest to many in the condensed matter community. I find the results convincing and I would expect the paper to be highly cited.

(1) Figure 2 shows the magnetic susceptibility. The authors say they measured under both FC and ZFC conditions but they do not say which data they have plotted. They are at pains to emphasise that various aspects of their results are inconsistent with the formation of a spin glass at the 1:1 composition and overlapping ZFC and FC curves would provide further evidence.

Answer: We agree that this should be clearly stated in the paper. The ZFC and FC curves overlap, and for this reason we have chosen to only show the ZFC curve. We have added a short remark to the main text and Fig. 2 caption.

(2) There is one fundamental point that I don't fully understand. I can see that the switch from a full to an empty d shell will modify the strength of the J_2 Cu - O - B' - O - Cu interaction and it will therefore change the J_1/J_2 ratio. However, the authors don't explain why Te vs W should cause a 6-fold change in the strength of the J_1 Cu - O - O - Cu interaction, which it apparently does. Any ideas?

Answer: This is an important point to make and is still an open question in the literature. Babkevich et al. [PRL 117 237203] propose the Cu - O - O - Cu pathway without any contribution

from Te^{6+} based on calculations on small clusters (CASCCF method). Xu et al. [JPCM 29 105801] propose that the unoccupied Te^{6+} 5p orbitals actually do hybridize to a small degree with O 2p orbitals, which could explain the increase in J1 compared to Sr_2CuWO_6 . This prediction is based on density functional theory calculations within the GGA+U framework.

REVIEWERS' COMMENTS:

Reviewer #1 (Remarks to the Author):

In the new version the authors have included new relevant low-temperature muSR results which do support their statements. Considering that they have satisfactorily complied to my remarks and improved the manuscript, I find that the manuscript is now suitable for publication.

Reviewer #2 (Remarks to the Author):

I am pleased with the revisions that have been made to this paper. The additional analysis of the heat capacity and muSR experiments (with additional data points) provides a more solid conclusion to their manuscript. It is remarkable that glassiness is not induced in this case via doping - which would have been seen in the muSR. I am convinced that spin liquid behaviour appears as a function of cation tuning.

I would endorse the publication of this manuscript as is.

Reviewer #3 (Remarks to the Author):

The authors have modified the manuscript in a way that provides satisfactory answers to the questions I posed in my previous review. In my opinion the paper can now go forward for publication.